# Developing a Flying Explorer for Autonomous Digital Modelling in Wild Unknowns

**DOI:** 10.3390/s24031021

**Published:** 2024-02-05

**Authors:** Naizhong Zhang, Yaoqiang Pan, Yangwen Jin, Peiqi Jin, Kewei Hu, Xiao Huang, Hanwen Kang

**Affiliations:** 1College of Civil Aviation, Nanjing University of Aeronautics and Astronautics, Nanjing 211106, China; zhangnaizhong@nuaa.edu.cn (N.Z.); huangxiao@nuaa.edu.cn (X.H.); 2College of Engineering, South China Agriculture University, Guangzhou 510070, China; 20213142022@stu.scau.edu.cn (Y.P.); yangtsejin@stu.scau.edu.cn (Y.J.); 20223173023@stu.scau.edu.cn (P.J.); huck_weeeee@whu.edu.cn (K.H.)

**Keywords:** flying robot, exploration, digital modelling, drone

## Abstract

Digital modelling stands as a pivotal step in the realm of Digital Twinning. The future trend of Digital Twinning involves automated exploration and environmental modelling in complex scenes. In our study, we propose an innovative solution for robot odometry, path planning, and exploration in unknown outdoor environments, with a focus on Digital modelling. The approach uses a minimum cost formulation with pseudo-randomly generated objectives, integrating multi-path planning and evaluation, with emphasis on full coverage of unknown maps based on feasible boundaries of interest. The approach allows for dynamic changes to expected targets and behaviours. The evaluation is conducted on a robotic platform with a lightweight 3D LiDAR sensor model. The robustness of different types of odometry is compared, and the impact of parameters on motion planning is explored. The consistency and efficiency of exploring completely unknown areas are assessed in both indoor and outdoor scenarios. The experiment shows that the method proposed in this article can complete autonomous exploration and environmental modelling tasks in complex indoor and outdoor scenes. Finally, the study concludes by summarizing the reasons for exploration failures and outlining future focuses in this domain.

## 1. Introduction

The Digital Twin (DT) concept, introduced by NASA in 1991, involves virtual representations of physical assets that reflect their context through data from various sensors [1]. Digital modelling (DM) is a critical task within the DT framework [1], contributing to insights, predictions and performance optimisation in industries such as manufacturing and transportation. Industries that rely on digital models, such as agriculture and construction, benefit greatly from drones with exceptional obstacle-avoidance capabilities [2]. These drones can generate 3D models of complex terrains and intricate plantations. The introduction of digital models in plantations significantly enhances daily tasks like harvest planning, and revolutionizing orchard management [3,4].

This study explores a unique mapping and exploration theory that paves the way for autonomous, drone-based, rapid exploration models of orchards. In the field of management, the development of autonomous robotic platforms with exploration and mapping capabilities is revolutionary. Robots, especially those equipped with 3D LiDAR sensors, contribute essential elements to DM, including accurate 3D modelling [3]. Their ability to mark characteristic points at specific locations is proving beneficial for later data collection and processing [5,6].

Exploring and mapping unknown and unstructured environments have always been an important and fundamental task for robots. This includes tasks such as inspecting large infrastructures, surveying buildings, performing search and rescue missions, and exploring underground spaces [7]. The exploration of wild, unknown and unstructured environments, especially those that are harsh and lack GPS connectivity, has become a major focus for autonomous robotic exploration missions. This focus has grown significantly in recent years, particularly with events such as the DARPA Subterranean Challenge. In this challenge, teams of robots are tasked with exploring and identifying artefacts in complex and unstructured environments. In conjunction with Simultaneous Localisation and Mapping (SLAM), overcoming the challenges of motion planning and exploration behaviour is one of the most fundamental issues in these diverse applications.

Robotic localisation, path planning, and exploration are fundamental research areas for building such a robotic system [8], with various solutions to the challenge of fully exploring unknown areas, including pattern-based approaches, frontier methods and entropy-based strategies [9]. Occupancy-based path planning, which relies on maps of occupied and free space, often uses algorithms such as Dijkstra’s or advanced versions of A* and jump-point-search. Alternatively, the Rapidly Exploring Random Tree (RRT) algorithm, in its modern versions, serves as a core component of some path planning solutions, offering computational efficiency and flexibility, but lacking the guarantee of finding the shortest path within a limited number of iterations.

Traditionally, exploration and planning problems have been treated separately. For example, in ref. [10], a deep reinforcement model selects optimal bounds and an A* algorithm finds the shortest path. In ref. [11], stochastic differential equations identify optimal bounds and an RRT* path planner computes the path. Recently, the integration of exploration into the central planning problem has gained attention, with solutions such as Next-Best-View planners [12] being foundational. Exploration RRT (ERRT) [13] shares core concepts with Next-Best-View planners but is innovative in its algorithmic implementation. ERRT explicitly solves paths to pseudo-random objectives, which distinguishes it from the iterative evaluation of RRT branches. In particular, ERRT computes optimal actuation for trajectory following through a receding horizon NMPC problem, with the proposed method using the Optimization Engine, an open-source Rust-based optimization software. In the context of unmanned aerial vehicles (UAVs), energy constraints impact the duration of flight operations and the spatial coverage range. Addressing this challenge, the PAD-based remote sensing (PBRS) path planning framework intervenes at the planning stage, facilitating extended UAV operations over vast areas [14].

This work presents a novel solution to the combined challenge of robotic odometry, path planning, and exploration for digital modelling in wild unknown environments. It employs a minimum cost formulation with pseudo-randomly generated objectives, integrating multi-path planning and evaluation. This evaluation considers information gain and total distance. Our approach emphasises full coverage of the unknown map, depending on the availability of feasible boundaries of interest. The evaluation focuses on a robotic platform with a lightweight 3D LiDAR sensor model and evaluates the consistency and efficiency of exploring a completely unknown subterranean-like and unstructured area. The algorithm is designed with versatile robot localisation and motion planning that allows dynamic changes to the desired goal and behaviour. Our detailed contributions are presented below.

Developing the Aerial Robot Exploration (AREX) system, as shown in Figure 1, including its hardware and software framework, for autonomous digital modelling in unstructured and unknown environments.Developing a novel algorithm for autonomous exploration in unknown environments for multiple scenarios.Demonstration of the developed AREX system in real-world applications, showcases its effectiveness in several scenarios.

## 2. Materials and Methods

### 2.1. Robot Design

#### 2.1.1. Hardware Design

AREX features a frame design that prioritises strength, durability and ease of assembly. Its rigid bodywork is made from carbon fibre with a density of 0.00153 ${\mathrm{mm}}^{3}$/g, resulting in a lightweight 250 g frame. The diagonal engine wheelbase is 380 mm, powered by a 4s 30C 4000 mAh battery. The platform features four SUNNYSKY X-2216 II KV880 (SunnySky, Columbus, OH, USA) brushless DC motors with APC 9045 (APC Technology Group, Rochester, UK) two-blade propellers, controlled by an XF 35A (Cyclone, Dongguan, China) 4in1 electronic speed controller (ESC). For autopilot functions, the Pixhack 6c mini (Holybro, HongKong, China) provides attitude and thrust control. The AREX measures 500 × 500 × 220 mm, weighs 1.9 kg and has an endurance of 10 min.

The schematic in Figure 2b illustrates AREX’s configuration. The Pixhack 6c Mini Flight Controller communicates with the ESC using the Dshot600 protocol, while also connecting to the on-board computer via a UART. This allows IMU data to be received from the onboard computer and desired attitude and throttle commands to be received from the flight plan. The sensor setup, shown in Figure 2, includes a LIVOX MID-360 (Livox Technology, Shenzhen, China) mixed LiDAR with a 360° horizontal and 59° vertical field of view. At the front, a RealSense D430 (Intel RealSense, Santa Clara, CA, USA) depth camera operates at 10 Hz for the LiDAR and 30 Hz for the camera. The Pixhack flight controller outputs 9-axis IMU data at 200 Hz.

#### 2.1.2. Software Design

The software architecture of AREX, shown in Figure 3, consists of several key modules. In the robot odometry module, data from LiDAR odometry or visual odometry is used to estimate the 6-axis Degree of Freedom (DOF) positional attitude of AREX. The resulting position and point cloud data are used in the exploration module to maintain an Octomap representing the exploration state of the environment. This module uses RRT to identify a series of boundary points in the unknown region of the Octomap. The directional RRT exploration proposed in this paper determines the final exploration target point. This information is then fed into the motion planning module which, using a PD controller, calculates the desired attitude and throttle for AREX based on the path points. Finally, the calculated commands are sent to the flight controller. In cases where the stop strategy is triggered, AREX returns autonomously to the starting point and lands or remains in place.

### 2.2. Robot Odometry

#### 2.2.1. Visual Odometry

VINS-Fusion [15] is used for visual odometry, and the loss function is constructed in the formula below:(1)$$\begin{array}{c} \min_{X}\left\{\underset{\mathrm{prior}\;\mathrm{factor}}{\underset{︸}{∥r_{p}-H_{p}{X∥}^{2}}}+\underset{\mathrm{IMU}\;\mathrm{propagation}\;\mathrm{factor}}{\underset{︸}{\sum_{k\in B}{∥r_{B}\left({\hat{z}}_{b_{k+1}}^{b_{k}},X\right)∥}_{P_{b_{k+1}}^{b_{k}}}^{2}}}\right.\\ \left.+\underset{\mathrm{vision}\;\mathrm{factor}}{\underset{︸}{\sum_{(l,j)\in C}\rho (∥r_{C}\left({\hat{z}}_{l}^{c_{j}},X\right){∥}_{P_{l}^{c_{j}}}^{2})}}\right\} \end{array}$$
where the first item is the priority after the marginalization, the second item is the IMU residual, and the third item is the visual residual. Because of the complex design, this system is very robust, so we combine it with improved FAST-LIO for better odometry. A graph is shown in Figure 4 to illustrate the framework of VINS-Fusion.

#### 2.2.2. LiDAR Odometry

FAST-LIO [16] is used for LiDAR odometry, and an error-state iterated Kalman filter (ESIKF) is used to update the state of the system. The overview of improved FAST-LIO is shown in Figure 5. The linearized update formula of the error-state variables in FAST-LIO is
(2)$${\tilde{x}}_{i+1}≃F_{\tilde{x}}{\tilde{x}}_{i}+F_{w}w_{i}$$
where the matrix $F_{\tilde{x}}$ and $F_{w}$ in Equation (Equation 2) is computed in Equation (Equation 3).



(3)
Fx˜=Exp−ωm−b^ωΔt00−Aωm−b^ωΔtTΔt000IIΔt000−R^IiGam−b^a∧Δt0I0−R^IiGΔtIΔt000I000000I000000I     ≈I00IΔt000IIΔt000−R^IiGam−b^a∧Δt0I0−R^IiGΔtIΔt000I000000I000000I Fw=−Aωm−b^ωΔtTΔt00000000−R^IiGΔt0000IΔt0000IΔt0000     ≈−ITΔt00000000−R^IiGΔt0000IΔt0000IΔt0000



We can obtain new state variables through forward propagation, which is used to compensate for the motion distortion of the LiDAR in backward propagation, the calculation formula is shown below:(4)pfjLk=TL−1ITˇIjIkTLIpfjLj

This LIO algorithm can quickly and robustly output LiDAR odometry, but the frequency of the odometry it outputs is only 10 Hz, which is too low for robots that need to obtain pose information in real-time. A high-frequency LiDAR inertial odometer is used here to achieve the frequency required by robot, whose frequency of odometry can reach 200 Hz. Whenever the ESKF is updated with a frame of odometry, this odometry will be recorded, and the newly incoming IMU measurement data will be integrated based on this optimized odometry. The calculation is shown in the following formulas. When a new frame of IMU data $a_{m}$ arrives:(5)$$a_{c}=-\frac{a_{m}}{∥a_{m}∥}\cdot G_{m}$$
where $a_{m}$ represents the acceleration measurement of IMU, $a_{c}$ represents the input acceleration of current time, $G_{m}$ represents the preset gravity constant, which is a scalar.

Since we record the optimized pose output by the system, we have:(6)$$a_{0}=R_{i}\left(a_{p}-b_{a}\right)-g$$
where $a_{p}$ represents the input acceleration of previous time, $R_{i}$ is the previous pose, $b_{a}$ is the bias of acceleration, g is the estimated gravity in odonetry of FAST-LIO. Then, using median angular velocity to update the rotation matrix:(7)$$\begin{array}{cc} \omega & =\frac{\omega_{0}+\omega_{1}}{2}\\ R_{i+1} & =R_{i}\mathrm{Exp}\left(\omega \Delta t\right) \end{array}$$
where $\omega_{0}$ is the previous angular velocity measurement of IMU, $\omega_{1}$ is the current angular velocity measurement of IMU. Then, transfer the acceleration to the world coordinate and calculate the median value:(8)$$\begin{array}{cc} a & =\frac{a_{0}+a_{1}}{2} \end{array}$$

Then, the position and linear velocity can be updated as follows:(9)$$\begin{array}{cc} p_{i+1} & =p_{i}+v_{i}\Delta t+\frac{1}{2}a\Delta t^{2}\\ v_{i+1} & =v_{i}+a\Delta t \end{array}$$

### 2.3. Robot Motion Planning

#### 2.3.1. Environment Representation

The motion planning map is a localised raster map composed of radar point cloud frames represented by a one-dimensional array. The value of each array element indicates the occupancy state of the corresponding grid, with 0 for free space and 1 for obstacles. The map has default dimensions of L, W, H and a grid size of $size_{grid}$, resulting in an array size of array_size. The calculation for array_size is given by
(10)$$\mathrm{array}\_\mathrm{size}=\frac{L\cdot W\cdot H}{size_{grid}^{3}}$$

The index value of each grid can be obtained by Equation (Equation 11)
(11)$$Id_{i}=h_{i}\cdot num_{w}\cdot num_{l}+w_{i}\cdot num_{l}+l_{i}$$
where $h_{i}$, $w_{i}$, $l_{i}$ represent the indices of layers in three directions of $grid_{i}$, $num_{w}$ denotes the number of grids in the width direction, and $num_{l}$ is similar to $num_{w}$.

On receipt of a point cloud frame, the 2D array values representing local map obstacle locations are set to zero. Each spatial point in the point cloud frame calculates the corresponding index value and assigns the value 1 to the corresponding position in the array. Spatial points outside the specified map area are discarded. The layer indexes of the raster in which the spatial points are located are as follows:(12)$$\begin{array}{cc} w_{i} & =\left\lfloor \frac{P_{i}.x}{size_{grid}}\right\rfloor +\frac{num_{w}}{2}\\ l_{i} & =\left\lfloor \frac{P_{i}.y}{size_{grid}}\right\rfloor +\frac{num_{l}}{2}\\ h_{i} & =\left\lfloor \frac{P_{i}.z}{size_{grid}}\right\rfloor +\frac{num_{h}}{2} \end{array}$$

To improve the feasibility of the motion planning path, obstacles are expanded by applying a preset expansion coefficient to each point in the point cloud frame, generating an expanded point set. The corresponding array positions are then assigned a value of 1 based on the spatial locations of the points in this set. The point set expression formula is as follows:(13)$$\begin{array}{c} \left[\left(P_{i}\cdot x-\lambda_{\mathrm{inflat}\;}\right)+{\mathrm{size}}_{\mathrm{grid}}\cdot a\right.\\ \left(P_{i}\cdot y-\lambda_{\mathrm{inflat}\;}\right)+{\mathrm{size}}_{\mathrm{grid}}\cdot b\\ {\left.\left(P_{i}.z-\lambda_{\mathrm{inflat}\;}\right)+{\mathrm{size}}_{\mathrm{grid}}\cdot c\right]}^{T}\\ a,b,c\in [0,n_{step}]\\ n_{step}=\left\lceil \frac{\lambda_{inflat}}{size_{grid}}\right\rceil \end{array}$$

#### 2.3.2. Local Motion Planner

Our work follows EGO-planner [17], an optimization-based framework for robot motion planning.

***Collision-Free Trajectory Generation:*** By giving a motion goal, the motion planning module generates an initial B-spline path Φ that does not take collisions into account. The start and end points of the path are the robot’s position and the goal, respectively. Then, the control point $Q_{i}$, which exists in the obstacle, finds the anchor point $p_{i}$ at the obstacle surface for the control point that exists inside the obstacle. Then, the obstacle distance from $Q_{i}$ to the obstacle is defined as
(14)$$\begin{array}{c} d_{i}=\left(Q_{i}-p_{ij}\right)\cdot v_{i} \end{array}$$A collision-free path is obtained by optimizing the position of $Q_{i}$ such that $d_{i}$ > 0.

***Grediant-based Trajectory Optimization***: The cost function of the trajectory optimization is:(15)$$\begin{array}{c} minJ=\lambda_{s}J_{s}+\lambda_{c}J_{c}+\lambda_{d}J_{d}, \end{array}$$
where $J_{s}$ is the smoothness penalty, $J_{c}$ is for collision, and $J_{d}$ indicates feasibility. $\lambda_{s}$, $\lambda_{c}$, $\lambda_{d}$ are weights for each penalty terms. We use a uniform B-spline, which means that for each control point $Q_{i}\in R^{3}$, they have the same time interval from the latter control point, i.e., $▵t=t_{i+1}-t_{i}$. Thus, the velocity $V_{i}$, acceleration $A_{i}$, and jerk $J_{i}$ at each control point are obtained by
(16)$$\begin{array}{c} V_{i}=\frac{Q_{i+1}-Q_{i}}{▵t},A_{i}=\frac{V_{i+1}-V_{i}}{▵t},J_{i}=\frac{A_{i+1}-A_{i}}{▵t}. \end{array}$$

For the smoothing term penalty, we examine the squares of the second-order and third-order derivatives of the control points concerning ▵t each. The second- and third-order derivatives of the B-spline curve are reduced by minimizing the second- and third-order derivatives of the control points for the node vector ▵t using the convex envelope property of the B-spline. This smoothness penalty is defined as follows, considering a total of *n* control points on the B-spline:(17)$$J_{s}=\sum_{i=1}^{n-1}∥A_{i}{∥}^{2}+\sum_{i=1}^{n-2}{∥J_{i}∥}^{2}$$

For the collision penalty, we set a safe distance $S_{f}$ and satisfy $d_{i}<d_{s}$ by the penalty requirement function. The collision penalty function is as follows:(18)$$\begin{array}{cc} j_{c}\left(i\right) & =\left\{\begin{array}{cc} 0 & \left(c_{i}\leq 0\right)\\ c_{i}^{3} & \left(0<c_{i}\leq s_{f}\right)\\ 3s_{f}c_{i}^{2}-3s_{f}^{2}c_{i}+s_{f}^{3} & \left(c_{i}>s_{f}\right) \end{array}\right.\\ where\;c_{i} & =s_{f}-d_{i}, \end{array}$$
where $d_{ij}$ comes froms Equation (Equation 14). Hence, the total collision penalties across the entire path result from the summation of penalties associated with individual control points $Q_{i}$. The formula can be expressed as follows:(19)$$\begin{array}{c} J_{c}=\sum_{i=1}^{n}j_{c}\left(Q_{i}\right). \end{array}$$

To guarantee feasibility in the context of the feasibility penalty, higher-order derivatives of each control in the *x*, *y*, and *z* dimensions are constrained to values below a predefined kinetic limit. The penalty function is articulated as follows:(20)$$\begin{array}{c} J_{d}=\sum_{i=1}^{n}w_{v}F\left(V_{i}\right)+\sum_{i=1}^{n-1}w_{a}F\left(A_{i}\right)+\sum_{i=1}^{n-2}w_{j}F\left(J_{i}\right) \end{array}$$

$w_{v},w_{a},$ and $w_{j}$ represent the weights assigned to different higher-order derivatives for each control point. Meanwhile, the definition of F(·) is as follows:(21)$$\begin{array}{c} F\left(C\right)=\sum_{r=x,y,z}f\left(c_{r}\right) \end{array}$$
(22)$$\begin{array}{c} f\left(c_{r}\right)=\left\{\begin{array}{cc} a_{1}c_{r}^{2}+b_{1}c_{r}+c_{1} & \left(c_{r}\leq -c_{j}\right)\\ {\left(-\lambda c_{m}-c_{r}\right)}^{3} & \left(-c_{j}<c_{r}<-\lambda c_{m}\right)\\ 0 & \left(-\lambda c_{m}\leq c_{r}\leq \lambda c_{m}\right)\\ {\left(c_{r}-\lambda c_{m}\right)}^{3} & \left(\lambda c_{m}<c_{r}<c_{j}\right)\\ a_{2}c_{r}^{2}+b_{2}c_{r}+c_{2} & \left(c_{r}\geq c_{j}\right) \end{array}\right. \end{array}$$
where $c_{r}\in C\in \left\{V_{i},A_{i},J_{i}\right\},a_{1},b_{1},c_{1},a_{2},b_{2},c_{2}$ satisfy second-order continuity. $c_{m}$ is the derivative limit, and $c_{j}$ represents the intersection points between the quadratic and cubic intervals. For a given condition where λ<1−ϵ and ϵ≪1, the objective is to ensure the structure adheres to the imposed constraint.

### 2.4. Robot Exploration Module

#### 2.4.1. Environmental Modelling

By maintaining the collection of undistorted LiDAR feature points, a high-precision global point cloud map is composed. This process is outlined by Equation (Equation 23). pfjLk represents the undistorted LiDAR feature points in the LiDAR coordinate obtained through (Equation 4), TLI denotes the transformation relationship from the LiDAR coordinate to the IMU coordinate, and TIkG signifies the system state corresponding to the LiDAR frame.
(23)p¯fjG=T¯IkIGTLpfjLk;j=1,⋯,m.

A 2D occupancy map is constructed to represent the known and unknown regions in the scene, where each raster has only three values: −1, 0, and 1 for the unknown, idle, and occupied states, respectively. When using the RRT-based boundary point detector, by querying whether the grid on the tree growth line is the unknown state value −1, we can determine whether to find the boundary point of the unknown region.

The occupancy state of the occupied map is updated by LiDAR point cloud frames in global coordinates, and we select the point clouds with height values of 0–2 m in the point cloud frames in global coordinates to represent the state of the raster. A grid is considered occupied when a point cloud exists in the grid, while a grid is considered idle when the occupied grid is connected to the robot’s position and the line passes through it. As the explorer moves, the LiDAR point cloud frames change with the environment, and the occupancy map is updated by aggregating the odometer information.

#### 2.4.2. Direction-Aware RRT Exploration

Algorithm 1 demonstrates the direction-aware exploration based on RRT. The following are the relevant definitions:R represents the data frames from the LiDAR.O represents the odometry information.M represents the grid map.CGs represents the frontier points.$T_{t}$ represents the expected exploration goal at time t.$P_{s}$ represents the the smoothed path.$E_{x}$ represents the flag for executing explorationThe four key nodes shown in Algorithm 1 are described below:Octomap_Server: accepts radar data frame information and odometer information for constructing a 2D occupancy map representing the scene exploration.Frontier_Detector: accepts the 2D occupancy map and odometer information constructed by Octomap Server and searches for boundary points of the unexplored environment by RRT and outputs a list of candidate target points.Revenue_Calculator: accepts the 2D occupancy map, odometer information, and candidate target points, calculates the value of the utility function of the candidate target points, and outputs the target point with the highest score.Path_Planner: accepts target points, radar data frames, and odometer information to plan a collision-free smooth path for the robot.
**Algorithm 1:** Direction-aware RRT exploration   
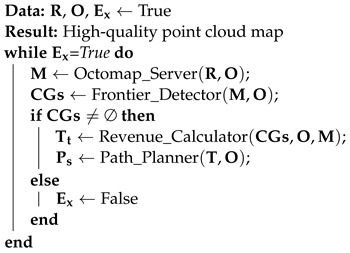



When the direction-aware RRT exploration algorithm receives data frames from LiDAR R, odometry O, and the flag for executing exploration $E_{x}$, the exploration task is activated. The Octomap_Server processes the data frames from LiDAR R and odometry O to build a grid map M. The Frontier_Detector receives the grid map M and searches for candidate target points CGs. When the set of candidate target points CGs is not empty, the Revenue_Calculator takes the candidate target points CGs, odometry O, and grid map M to calculate the expected exploration goal $T_{t}$. Subsequently, the Path_Planner plans the path $P_{s}$ and controls the UAV to reach the expected exploration goal $T_{t}$. If the Frontier_Detector fails to find candidate target points CGs, the flag for executing exploration $E_{x}$ is set to false. Alternatively, if the revenue of target points in the Revenue_Calculator falls below a threshold, the flag for executing exploration $E_{x}$ is also set to false. Exploration stops when the flag for executing exploration is false.

#### 2.4.3. Evaluation Strategy

The exploration strategy in this work evaluates candidate observation positions to find the optimal frontier vertex. This strategic approach aims to shorten the exploration time, reduce the overall step size, and ultimately increase efficiency [18,19,20]. Our strategy takes into account three key factors: the distance of the robot to the observation position, the expected information gain, and the correlation with the direction of the last exploration target. Figure 6 visually demonstrates our exploration strategy.
(24)$$\begin{array}{cc} R(CG_{i}) & =\lambda_{I}\cdot f_{i}^{I}-\lambda_{N}\cdot f_{i}^{N}\\ CG_{i} & =\mathrm{Arg}\max_{CG_{i}\in CGs}\left(R(CG_{i})\right) \end{array}$$

In the explored method [21], the robot chooses the nearest, accessible and unvisited border with the largest grid size as its goal. The choice of the desired border is made by maximising the utility function (Equation 24), where $f_{i}^{I}$ and $f_{i}^{N}$ represent the grid size of the border area and the spatial distance between the robot and the border node, respectively. $\lambda_{I}$ and $\lambda_{N}$ are weighting parameters associated with these two terms. Importantly, the utility function reaches its maximum value when the frontier size is large and the distance is small. This optimisation allows the robot to navigate efficiently along the shortest obstacle-free path from its current cell to the cell containing the map coordinate. As emphasised in [22], the consistent movement towards new frontiers allows the robot to extend its map into unexplored areas until the entire environment is covered.

In unstructured environments, the existing boundary selection method may become inefficient [23] because robots tend to prioritise boundary points with high local information returns [24]. As a result, the robot may be attracted to these points, disregarding the continuity of environmental information in the current direction of exploration. This leads to the robot revisiting previously explored locations when moving towards points with lower information returns [25]. To address this inefficiency, this paper improves the exploration strategy by incorporating a term influenced by the direction of the exploration target point into the utility function for profit calculation. The improved utility function $R_{d}\left(CG_{i}\right)$ is expressed as follows
(25)$$\begin{array}{cc} R_{d}\left(CG_{i}\right) & =\lambda_{I}\cdot f_{i}^{I}-\lambda_{N}\cdot f_{i}^{N}-f_{i}^{D}\\ f_{i}^{D} & =e^{\lambda_{D}\cdot A} \end{array}$$
where $f_{i}^{I}$, $f_{i}^{N}$ are defined the same as Equation (Equation 24). $f_{i}^{D}$ is an exponential function of *A*. *A* is the angle between the vectors $\vec{T_{t-2}T_{t-1}}$ and $\vec{T_{t-1}CG_{i}}$. Similarly, $\lambda_{I}$, $\lambda_{N}$, and $\lambda_{D}$ are user-defined weightings for the size, distance and exploration direction.

#### 2.4.4. Stop Cretria

The termination criterion for exploration is a critical aspect of the exploration process described above [26]. In practical scenarios, it is often unrealistic to set a specific percentage of exploration coverage in a given area as the termination point [27]. For flying robots, efficient management of their power supply is essential due to energy constraints [28]. It is therefore necessary to consider stopping the exploration mission either when a particular area is completed or when the exploration yield becomes relatively low [29]. This stopping strategy is implemented to ensure that the robot can save sufficient power for a safe return. The exploration termination strategies used in this work consist of the following two criteria:No further candidate exploration target points are discovered. As shown below:
(26)CGs=∅The unknown area of the candidate region is below a preset threshold. As shown below, where $S_{u}$ represents the area of the unknown region near the corresponding target point, and *s* is the predefined threshold:
(27)$$S_{u}<s$$These two criteria exist side by side, and as long as either of these conditions is met, the exploration mission stops and the robot will return to the starting point.

## 3. Results

### 3.1. Experiment Setups

In real-world scenarios, we conducted experiments to validate the effectiveness of our work using the physical platform depicted in Figure 2. Our algorithm functions within the ROS Noetic operating system. We employ a method utilizing a PD controller to steer the robot along the planned path at a frequency of 100 Hz. Simultaneously, it detects potential collisions along the path at a rate of 20 Hz, triggering a rerouting of the current trajectory in the event of a collision. The global map resolution is configured at 0.1 m, while the dimensions of the local sensing map for path planning are set to 30 m × 30 m × 3 m. Additionally, the obstacle point cloud’s expansion radius is set to 0.5 m. Notably, the starting point is established as the map’s origin, denoted as $x_{\mathrm{start}}={\left[0,0\right]}^{T}$. The experimental setup consists of two different scenes: a regularly maintained forest and an underground parking lot.

### 3.2. Evaluation on Subsystems

#### 3.2.1. Comparison of Visual and LiDAR Odometry

In this section, we evaluate the performance of visual and LiDAR odometry in real-world environments, as shown in Figure 7. In this experiment, we utilized the robot kit illustrated in Figure 2a to conduct a comparative analysis between the visual and radar odometry systems. Our paper implements the improved FAST-LIO as the robot’s primary odometry method, concurrently running VINS-Fusion. Post-initialization, our motion planner, operated via Nomechine (8.11.3) software’s remote desktop, orchestrated the robot’s autonomous flight, adhering to six pre-set waypoints. Notably, the sixth waypoint serves as the starting point $x_{\mathrm{start}}={\left[0,0\right]}^{T}$. The outcomes are depicted in Figure 7.

From Figure 7a, the completion of initialization showcases an excellent alignment between the FAST-LIO and VINS-Fusion odometry. However, upon completing the mission involving six waypoints, Figure 7b distinctly illustrates a considerable shift in the VINS-Fusion odometry. In contrast, the LiDAR odometry remains stable, showing no significant deviation.

#### 3.2.2. Evaluation on Robot Motion Planning

We validate the utilised motion planning methods in the wild environments. In our experiments, we observed that the single-frame LiDAR point cloud output from the MID-360 tends to exhibit sparsity, potentially leading to certain smaller obstacles being unnoticed within the point cloud data. To address this, we amalgamated the multi-frame LiDAR point cloud data with that of the single-frame camera, presenting a comparative analysis in Figure 8a–d. Observing Figure 8a,b, it’s evident that the absence of an obstacle point cloud leads to an inadequate grid map representation, consequently rendering the planned path infeasible. However, through the fusion of 5-frame LiDAR point cloud data with single-frame camera data, depicted in Figure 8c,d, we effectively enhance our ability to perceive obstacles. The fusion of multiple sensors and frames of point cloud data addresses the challenge of perceiving smaller obstacles, enabling the motion planning process to generate feasible paths.

The motion planning for obstacle avoidance is based on optimizing paths that avoid inflated obstacles. Consequently, the choice of expansion coefficient size significantly impacts the success rate of obstacle avoidance. In Figure 8e–h, we showcase various obstacle avoidance paths for the same scene, each utilizing different expansion coefficients. In Figure 8e,f, the depicted obstacle avoidance paths utilize an expansion factor of 0.1. From Figure 8e, it’s noticeable that while these paths circumvent the expanding obstacles, they often appear excessively close to the obstacle point clouds in the original point cloud maps. Consequently, this proximity can potentially result in collisions due to the robot’s size or biases within the point clouds during actual flight.

Contrastingly, in Figure 8g,h, where the paths incorporate an expansion coefficient of 0.5, Figure 8h distinctly illustrates how an increased expansion coefficient effectively maintains a safer distance between the avoidance paths and the obstacles. This augmentation notably enhances the navigability and safety of the paths. However, the expansion coefficient is not “the bigger the better”; too high an expansion coefficient will cause the on-board computer to mistakenly think that it is in the obstacle and stop planning the path. Figure 8i,j demonstrates the on-board computer’s misjudgment in the case of expansion coefficient of 0.75.

### 3.3. Field Experiments

To verify the feasibility and robustness of the proposed system, we conducted exploratory experiments in two different scenarios.

#### 3.3.1. Scene 1

The first scenario was conducted in a forest located at South China Agricultural University in Guangzhou, Guangdong, China. After multiple tests, the coefficient $\lambda_{D}$ in Equation (Equation 25) for considering direction has been set to 0.35. The UAV demonstrates good performance when $\lambda_{D}$ is in the range of 0.32 to 0.45, maintaining exploration direction. The coefficient *s* in Equation (Equation 27) in the stopping criteria is set to 7.85 $m^{2}$, representing 10% of the LiDAR sensing range set in Octomap_Server. The exploration process is shown in Figure 9. According to the distribution range of the forest, we set the exploration boundaries centred on the starting point, with the farthest front/back point being 25 m, and the farthest left/right point being 20 m. We start the on-board computer through the remote desktop function of Nomechine to make the robot perform the autonomous exploration task. During the whole exploration process, the exploration module releases a total of six exploration target locations as shown in Figure 9a, which are provided with odometer information by the improved FAST-LIO in this paper, while the motion planning module used in this paper is responsible for local path planning and optimization. It is worth noting that the sixth exploration target point is the starting point, which is due to the stopping strategy triggered by the exploration module, so the coordinates of the starting point are released as the last target point so that the robot will return to the flight path and land on its own to complete the exploration mission. A semantic processing point cloud map based on our previous work [30], which can process object detection on point cloud in a Birds-Eye View (BEV), is utilised here to perform scene understanding in world environments, as shown in Figure 9b.

#### 3.3.2. Scene 2

The second scene took place in the underground parking garage of South China Agricultural University in Guangzhou, Guangdong, China, as depicted in Figure 10a. Utilizing the garage entrance as our starting point, we established an exploration boundary extending 100 m along the x-axis and 50 m along the y-axis from the starting point. Given the initial point’s orientation towards a 30-degree downhill slope, point cloud data ranging from −5.0 to −10 m in the z-axis direction was selected as the source for the 2D Octomap. The settings for other parameters remain consistent with the experiments in scene 1. To facilitate autonomous exploration, the onboard computer was activated using Nomechine’s remote desktop function, enabling the robot to execute exploration tasks. The actual exploration route is depicted in Figure 10a, while the exploration results are illustrated in Figure 10b. It is important to note that due to insufficient remaining battery power for the robot to return to the starting point, remote commands were dispatched to initiate an autonomous landing upon completing the exploration mission.

### 3.4. Failure Analysis

#### 3.4.1. Failure on Robot Odometry

In our experiments, odometer failure primarily occurs in two scenarios: (1) When the fuselage lacks sufficient structural strength, the flight control IMU (Inertial Measurement Unit) receives mechanical vibrations from the motor’s rotation. Consequently, the flight control system adjusts the motor based on the IMU waveform data, leading to resonance with the motor-transmitted vibration. (2) Odometer tracking failure arises during high-speed robot motion, especially when the robot undergoes rapid YAW angle changes. During exploration missions, when the target point significantly deviates from the robot’s current orientation, the PD controller principle prompts the robot to perform significant spins for orientation adjustment, disabling the odometer. To mitigate this issue, the direction-focused exploration strategy proposed in this paper circumvents such situations.

#### 3.4.2. Failure on Motion Planning

This failure manifests in two scenarios:Being entrapped within an obstacle group causes program judgment, leading to collision and subsequent planning failure. This behaviour stems from environmental perception inaccuracies due to LiDAR point cloud sparsity and measurement errors. Notably, the obstacle point cloud’s inconsistency, especially at object edges, causes sporadic jumps, resulting in the robot colliding with the obstacle cluster.Another prevalent failure involves obstacle avoidance. While achieving collision-free trajectories entails bypassing expanded obstacles in flight, determining an appropriate expansion coefficient in proportion to the airframe size is crucial. Excessively high expansion coefficients exacerbate the point cloud’s jumping phenomenon, leading to severe consequences.

Both issues stem from the inherent lack of accuracy in the employed environmental perception method. Addressing these challenges necessitates the development of a more precise environmental perception model.

## 4. Discussion

We conducted various experiments to compare the accuracy of different types of odometry, the performance of motion planning under different parameters, and the effectiveness of our approach in indoor and outdoor scenarios. We also analyzed the reasons for task failure during the exploration process. From Figure 7, it can be observed that, for the same motion path, the trajectory of LiDAR odometry can return to the starting point, while the visual odometry exhibits noticeable drift. This indicates that LiDAR odometry is more suitable as a UAV positioning module.

In the motion planning parameter tests, we achieved more accurate environmental perception by merging multiple frames of LiDAR point clouds. We compared the obstacle avoidance performance with different obstacle inflation parameters and found that the inflation factor needs to be set based on the size of the UAV. Both too-high or too-low inflation parameters can lead to planning failures.

We tested our proposed direction-aware exploration strategy in both outdoor and indoor scenarios. From Figure 9 and Figure 10, it can be seen that the direction-aware exploration strategy caused the UAV to explore in a specific direction and switch to a new direction only after reaching a boundary. After the UAV finished exploring, it triggered the stopping strategy to land automatically.

We analyzed and summarized the reasons for odometry failure in the experiments, identifying insufficient frame strength and significant spinning in planning as causes. However, the proposed direction-aware exploration strategy can avoid excessive spinning. Finally, our analysis revealed that motion planning failures are attributed to the inaccuracy of the environmental perception method, emphasizing the crucial role of precise environmental perception as a prerequisite for successful motion planning.

## 5. Conclusions

Our work introduced the design of AREX and its autonomous exploration capabilities. The system was equipped with robust localization, mapping, and efficient exploration target selection capabilities. In particular, it prioritized continuity in the exploration direction and minimized abrupt changes to increase efficiency and mitigate odometry errors. The global point cloud map generated by the exploration serves as a valuable resource for extracting semantic information about the environment. In our experiments, the real-time, high-precision environmental perception module proved crucial. The accuracy of environmental perception significantly influenced the identification of exploration boundaries and the effectiveness of obstacle avoidance. Future efforts will focus on developing more efficient and accurate methods for perceiving and representing the environment, as well as exploring robust path-planning approaches.

## Figures and Tables

**Figure 1 sensors-24-01021-f001:**
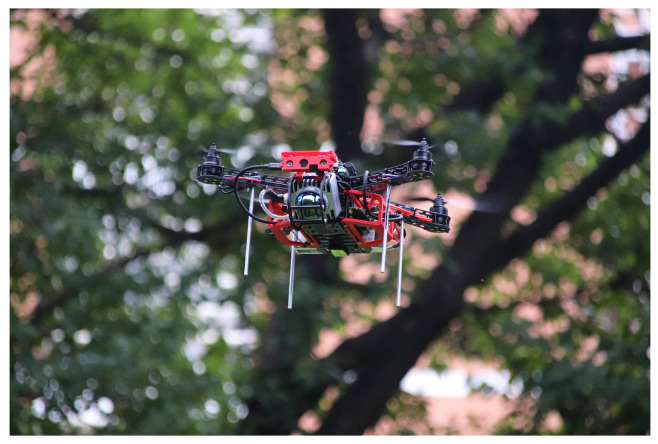
AREX: Flight system designed for high durability, multi-scenario exploration, and environmental modelling.

**Figure 2 sensors-24-01021-f002:**
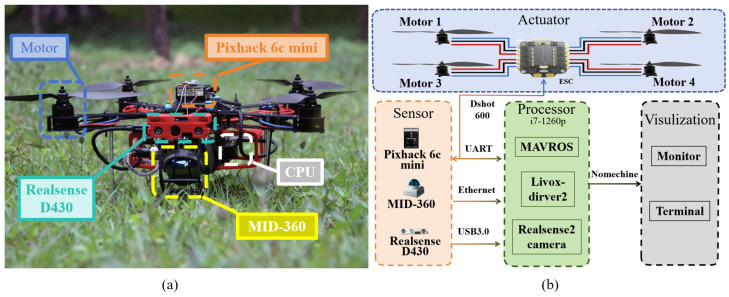
Robot design: (**a**) AREX’s frame structure and sensor layout. The MID-360 LiDAR is positioned at the front of the fuselage at a 45° angle to the horizontal z-axis, while the Realsense D430 is mounted above the MID-360 to fill in blind spots in the point cloud. The front dual arms of AREX form a 120° angle, maximizing the avoidance of obstructing the LIDAR’s scanning range. (**b**) AREX’s circuitry structure. Reflects the circuitry between the ESCs and motors, as well as the communication interfaces and protocols among various modules.

**Figure 3 sensors-24-01021-f003:**
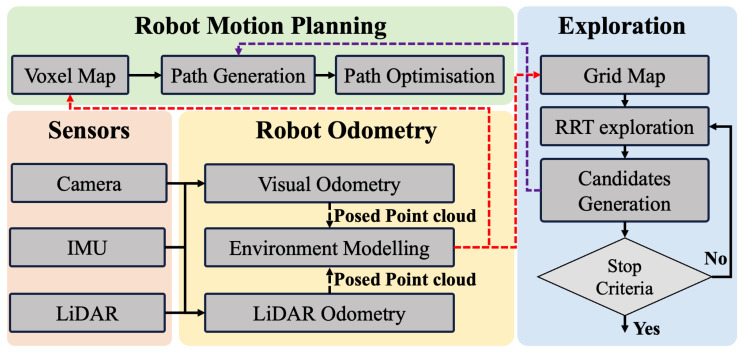
Software architecture of AREX.

**Figure 4 sensors-24-01021-f004:**
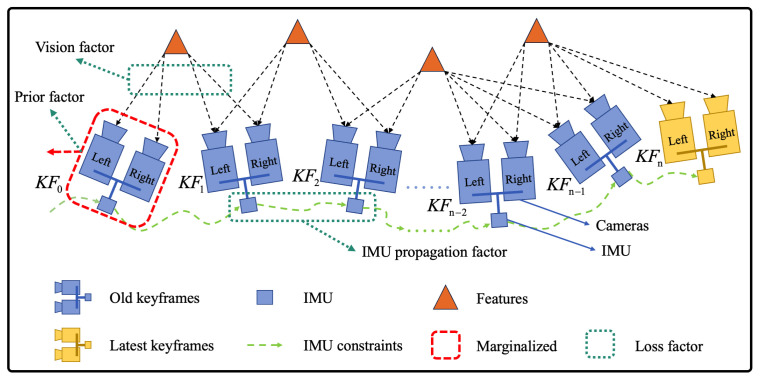
A graphic illustration of the VINS-Fusion framework. If the latest keyframe comes, it will be kept, and the visual and IMU measurements of the oldest frame will be marginalized. The prior factor of the loss function is obtained from marginalization. We can obtain the IMU propogation factor from IMU pre-integration. By computing the reprojection error between two keyframes, we can obtain the vision factor.

**Figure 5 sensors-24-01021-f005:**
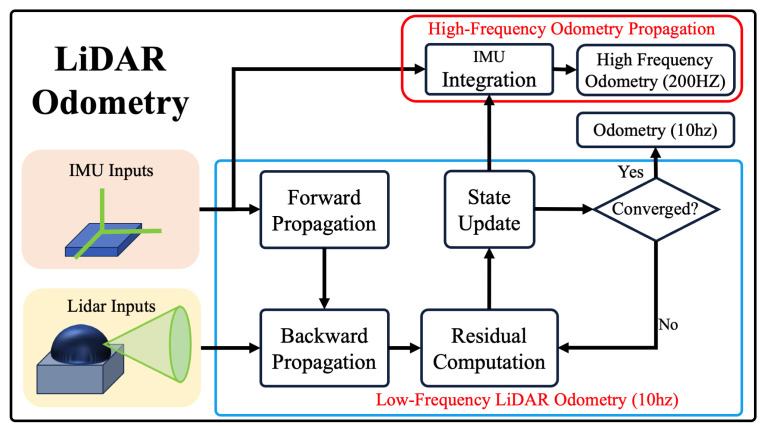
System overview of improved FAST-LIO, which can output high-frequency odometry.

**Figure 6 sensors-24-01021-f006:**
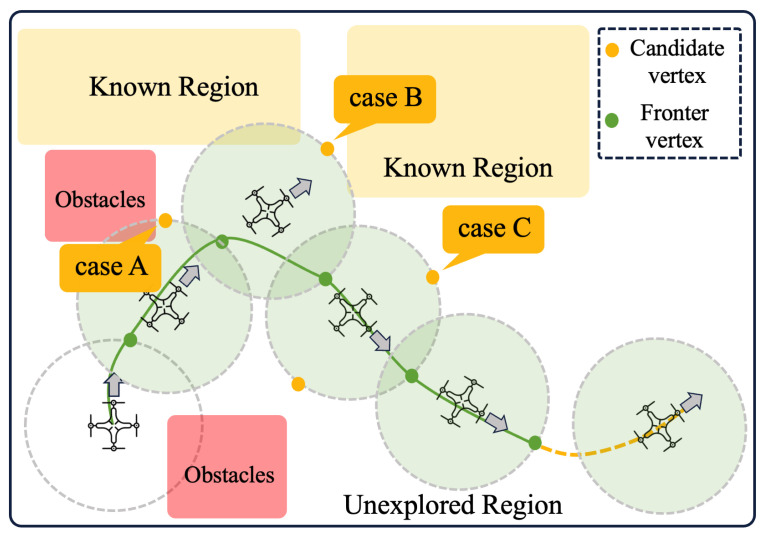
Demostration of RRT exploration strategy. Case A: Candidate vertexs close to an obstacle will not be considered. Case B: Candidate vertexes close to a known area will not be considered. Case C: Candidate vertex far away from the exploration direction will not be considered.

**Figure 7 sensors-24-01021-f007:**
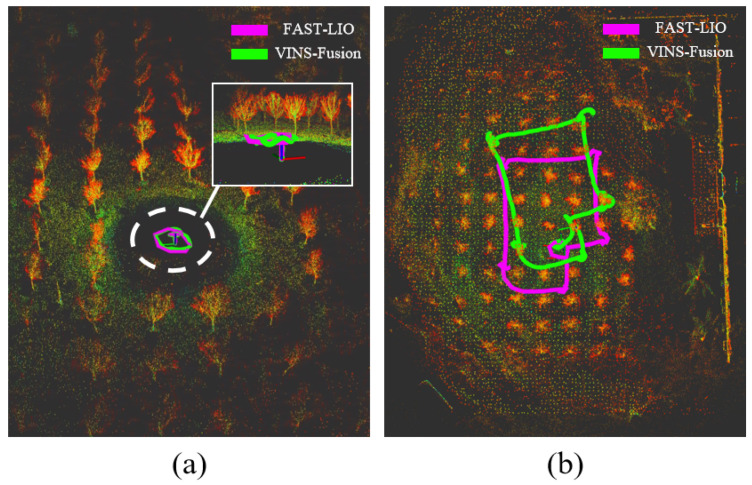
Comparison of visual and LiDAR odometry: (**a**) completion of odometry initialization; (**b**) comparison of global paths.

**Figure 8 sensors-24-01021-f008:**
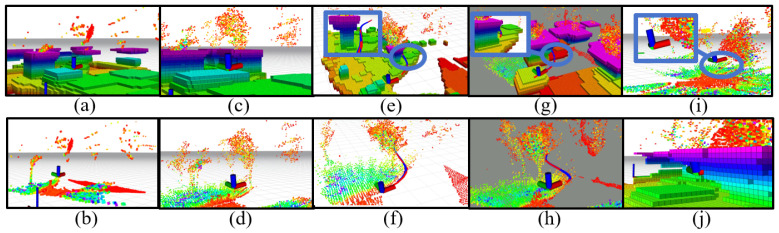
Effects of different parameters on motion planning. (**a**) Single-frame LiDAR point cloud expansion grid map. (**b**) Single-frame LiDAR raw point cloud. (**c**) Inflation grid map corresponding to the merged point cloud. (**d**) Merged raw point cloud. (**e**) Obstacle avoidance demonstration with 0.1 inflation coefficient. (**g**) Obstacle avoidance with a 0.5 inflation coefficient. (**f**,**h**) The obstacle avoidance paths of (**e**), and (**g**) in the raw point cloud. (**i**) Robot proximity to the obstacle point cloud. (**j**) robot within the inflated obstacle point cloud.

**Figure 9 sensors-24-01021-f009:**
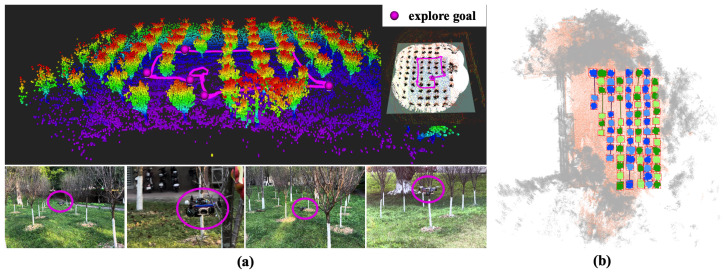
Instances of an autonomous exploration mission within the forest. (**a**) The exploration results and on-site flight demonstration in the forest. (**b**) Forest semantic map based on exploration results.

**Figure 10 sensors-24-01021-f010:**
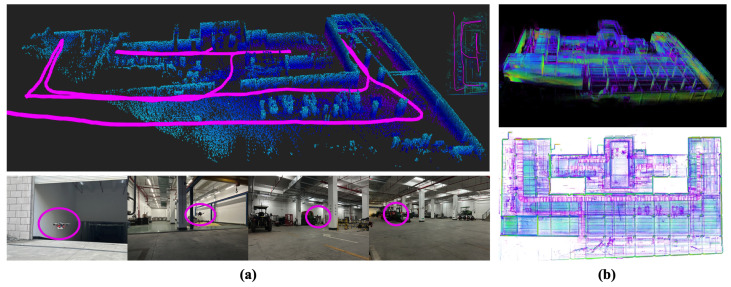
Presentation of the underground parking garage exploration process. (**a**) The exploration results and on-site flight demonstration in the underground parking garage. (**b**) Display of the global point cloud map in the underground parking garage.

## Data Availability

The data presented in this study are available on request from the corresponding author.

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
