# Peer review of "Developing a Flying Explorer for Autonomous Digital Modelling in Wild Unknowns"

_sensors, 2024, doi:10.3390/s24031021_

Round 1

Reviewer 1 Report

Comments and Suggestions for Authors

In this MS, the researchers introduced an innovative solution for robotic exploration in wild unknown environments, focusing on digital modeling. Their approach utilizes a minimum cost formulation with pseudo-randomly generated objectives, integrating multi-path planning and evaluation. The goal is to achieve full coverage of unknown maps based on feasible boundaries of interest, specifically emphasizing dynamic changes to the desired target and behavior. The research evaluated the algorithm on a robotic platform equipped with a lightweight 3D LiDAR sensor, assessing its consistency and efficiency in exploring completely unknown subterranean-like areas. The paper details the design of AREX, the robotic platform used, highlighting its robust localization, mapping, and efficient exploration target selection capabilities. Notably, the algorithm allows for dynamic adjustments to the exploration direction, aiming to increase efficiency and reduce odometry errors. The research emphasizes the critical role of a real-time, high-precision environmental perception module for accurate obstacle avoidance and exploration boundary identification.

Overall, the flow and the presentation of the paper are very well structured and the methodology is well presented. However, I have some minor suggestions that I have highlighted in the attached file.

Reviewer 2 Report

Comments and Suggestions for Authors

See the details in the attached file.

Reviewer 3 Report

Comments and Suggestions for Authors

Dear Authors,

Your study is interesting and of interest to readership of Sensors.

However, it needs some more improvement before being published.

The main concern for me is the lack of a general discussion section, which discusses your findings implications with other practices in the filed at a global perspective. You must convince the reader, why your findings are useful and important to the global reader. 

You may kindly find some other minor suggestions in the attached PDF file.

Looking forward to the revised version of the manuscript.

Kind Regards,

Comments on the Quality of English Language

English language is understandable, yet, a final proof reading is compulsory.

Please avoid passive voice!
